# Review of Development and Recent Advances in Biomedical X-ray Fluorescence Imaging

**DOI:** 10.3390/ijms241310990

**Published:** 2023-07-01

**Authors:** Theresa Staufer, Florian Grüner

**Affiliations:** Fachbereich Physik, Universität Hamburg and Center for Free-Electron Laser Science (CFEL), Luruper Chaussee 149, 22761 Hamburg, Germany; florian.gruener@uni-hamburg.de

**Keywords:** X-ray fluorescence imaging, preclinical applications, background reduction

## Abstract

The use of X-rays for non-invasive imaging has a long history, which has resulted in several well-established methods in preclinical as well as clinical applications, such as tomographic imaging or computed tomography. While projection radiography provides anatomical information, X-ray fluorescence analysis allows quantitative mapping of different elements in samples of interest. Typical applications so far comprise the identification and quantification of different elements and are mostly located in material sciences, archeology and environmental sciences, whereas the use of the technique in life sciences has been strongly limited by intrinsic spectral background issues arising in larger objects, so far. This background arises from multiple Compton-scattering events in the objects of interest and strongly limits the achievable minimum detectable marker concentrations. Here, we review the history and report on the recent promising developments of X-ray fluorescence imaging (XFI) in preclinical applications, and provide an outlook on the clinical translation of the technique, which can be realized by reducing the above-mentioned intrinsic background with dedicated algorithms and by novel X-ray sources.

## 1. Introduction

Since the famous discovery of X-rays by Wilhelm Conrad Röntgen in 1895 [1], various modalities of X-ray imaging have been developed, which all share the common feature of providing insights into organisms in a non-invasive manner. The basic principle of X-ray radiography was introduced into clinical practice in 1896, demonstrating the potential of X-ray imaging even for soft tissue by injecting a contrast medium [1]. In parallel to the rapid introduction of novel applications, many physicists worked hard on improving the basic X-ray technology, e.g., by improving the focusing of electron beams or optimizing the quality of the fluorescing screens to capture images of better quality [1]. While most of these early imaging methods mostly relied on the fact that the different transmission of X-rays in structures of varying density creates contrast on an imaging screen, Max von Laue discovered the principle of X-ray diffraction by crystals in 1912 [2]. After the confirmation of the diffraction discovery by William Henry Bragg and William Lawrence Bragg, a father and son, with an alternative method, two new research fields were born, X-ray crystallography and X-ray spectroscopy [2]. X-ray crystallography has since matured to one of the key methods in solving the three-dimensional structure of proteins, information which is essential to unravel the molecular function of proteins [3]. Nevertheless, several obstacles are still associated with the structural determination of proteins, such as the necessity to obtain large and well-diffracting crystals and the instability of purified proteins, which can be overcome by new techniques and strategies that have been developed over the past decade [3]. Besides crystallography, X-ray spectroscopy has also become an essential tool for obtaining local constituents’ data, in particular to identify features observed in imaging systems [4]. Nowadays, the sophistication of large-scale synchrotron beamlines capable of both soft and hard X-ray spectroscopy in recent years provides a wide variety of experiments that combine X-ray absorption (XAS) and X-ray emission (XES) to obtain detailed electronic structural information from a given absorber [5].

Besides techniques and applications using X-rays to study the smallest structures, radiological techniques were also gradually improved over the years, such as the now widespread computer tomography [1]. X-ray computed tomography (CT) consists of measuring attenuation profiles of transverse slices of patients from many different angular positions by using a fan or cone beam from an X-ray tube, in conjunction with a detector array traveling on a circular path opposite the X-ray source around a patient [6]. CT scans are now typically used to diagnose many diseases, such as various types of cancers, heart diseases such as myocardial disease and analyses of the liver or pancreas of patients [7]. Since the 1970s and 1980s, the speed of image acquisition has substantially improved, and modern CT scanners are capable of imaging patients in a matter of seconds or less [8]. A major drawback of CT is that large masses within the gastrointestinal tract may not be visible during scans, and more sophisticated methods such as dual-energy CT are required to differentiate materials with the same attenuation at a certain energy for better lesion depiction [7,9].

X-ray fluorescence (XRF) spectrometry, on the other hand, is based on the principle that individual atoms emit characteristic X-ray photons upon excitation by an external energy source. The abbreviations XRF and XFI are often used interchangeably, hence it is noted here that both versions will be used in the following, where the choice depends on the use of the cited publication. Different to other molecular imaging methods, the spatial resolution in XFI only depends on the size of the applied X-ray beam and does not face any physical limitations [10]. In order to make entities of interest visible with XFI, dedicated markers have to be coupled to them, such as metallic nanoparticles or molecular tracers, for example, iodine atoms. Considering that these markers do not decay over time, longitudinal studies are possible to study the biodistribution of labeled entities over long timespans in one and the same object. Furthermore, several entities can be tracked simultaneously by using different marker elements and measurements on completely different size scales, starting from full-body in vivo scans of small animals to measurements at the singe-cell level, which are feasible with XFI [11,12]. Especially the last two aspects are an advantage compared to other commonly used imaging methods, such as positron emission tomography (PET), where only single markers can be imaged over limited timespans due to the half-life of only 110 min of ^18^F, the workhorse of PET [13].

Compared to XFI, the sensitivity of CT is reduced, for instance, a tumor marked with gold nanoparticles in a tumor-bearing mouse model is undetectable for CT, while XFI can clearly locate it [12], as described in detail below. The main reason for this difference in the detection sensitivity is the fact that the contrast in CT arises due to a difference in photon counts in the forward (transmission) direction, while XFI is a spectroscopic method, where the fluorescence photons are emitted isotropically and the detection limit only depends on the spectral background in the signal region determined by multiple Compton scattering.

Glocker and Schreiber were the first to perform quantitative analysis of materials using XRF in 1928; however, only in the 1950s did the first commercially produced X-ray spectrometers become available, making the technique practicable [14]. Since then, improvements on the source, as well as on the detector side, have resulted in modern benchtop and handheld XRF systems, which are nowadays used in a variety of disciplines, such as forensic science, pharmaceuticals, cultural heritage and many others.

This review article aims to provide a historical overview of the development of X-ray fluorescence measurement techniques and to summarize the recent developments in the field. A detailed description of different experimental setups, X-ray sources used and the thereby achievable detection limits in various application areas are presented. As well as the current status in preclinical research, a translation of XFI to clinical applications is presented.

## 2. History and Developments of X-ray Fluorescence Measurement Techniques

In 1968, the first measurements of iodine in vivo in humans using an XRF setup were performed, resulting in a high concentration of about 400 μg/g in the thyroid gland [15]. Since the 1970s, the techniques have been improved to also allow determination of cadmium, lead, mercury, platinum and gold, with the main focus of connecting metal element abundance with surveillance of heavily exposed workers [16]. The in vivo application of X-ray fluorescence analysis was found to be limited to elements with atomic numbers larger than about 40 in 1980 [17] due to the absorption of the emitted characteristic radiation within the object. In those first in vivo measurement setups, different radiation sources, radionuclide sources and X-ray tubes were compared, and the emitted radiation was detected with Ge (Li)-detectors in combination with a collimator in front of the detector [17]. Recorded spectra of X-ray fluorescence measurements of lead in water when using ^57^Co sources showed a high background level, mostly caused by multiple scattered primary photons, and thereby showed limits in the minimum detectable concentration [17]. Similar minimum detectable concentration values could be achieved with X-ray tubes in combination with dedicated filters. However, those reported studies were carried out in human fingerbones, and it is stated in [17] that it is not possible to carry out detailed studies of the distribution of lead in the skeletons of occupationally exposed persons by means of X-ray fluorescence measurements in vivo because of the low sensitivity in measurements of deep-lying bones. Instead, X-ray fluorescence analysis is suggested to be used on autopsy samples, and hence only in an invasive way. The minimal detectable concentration of cadmium was studied by using a kidney model placed in water, with the conclusion that the sensitivity is very dependent on the layer of tissue between the detector and the kidney surface, and that the measurements mainly reflect the concentration in the kidney cortex instead of the whole organ [17].

An improved technique presented in [16] uses partly polarized photons and a detection angle of 90° to achieve a minimum background. A modified X-ray therapy tube is used in combination with rods and foils in order to make the scattered beam more monoenergetic and to also reduce the absorbed dose to the person sitting on a specifically designed chair. Kidney localization is performed using an ultrasound prior to fluorescence measurements. Collimated detectors containing thick sensors of either Si (Li) or Ge point at right angles to both the primary and the scattered beam, with a goal to reduce the number of background counts. Optimization strategies in [8] mainly consist of increasing the measurement time and using an X-ray tube, which mainly produces characteristic radiation of a high fluence rate to further decrease the detection limit.

A detailed discussion regarding the choice of the X-ray source, geometry and measurement sensitivity is presented in [18]. There, three main factors which affect the choice of the photon source are listed, namely, the need to maximize the lead X-ray fluorescence yield per incident photon, the adequate penetration depth, as well as minimizing the spectral background in the lead signal region. Compton-scattered photons are identified as the main source of background; hence, it is important to have the Compton scatter peak as far as possible from the lead X-ray peaks of interest [18]. ^109^Cd is identified as the source with the best parameters, together with a special collimator design to optimize the field of view and thereby reduce unnecessary doses to the subject and minimize the energy range and intensity of the detected Compton-scattered photons [18]. The estimations presented in [18] are based on the assumption that Compton scattering is isotropic in the laboratory system, and they suggest normalizing the detected lead counts to the coherent scatter peak (i.e., from Rayleigh scattering) in order to compensate for variations in object size and shape and in overlying tissue thickness. Besides measurements of lead concentrations in exposed workers, cisplatin, a cytostatic agent which has been proven to be successful in the treatment of malignant tumors, was followed in vivo by means of X-ray fluorescence [19]. In this study, a measurement setup for plane-polarized photons, in which the primary beam is scattered in two mutually orthogonal directions at a target with a low atomic number but a high density, is used to reduce the background contribution from incoherently scattered (Compton) photons to about 40%, compared to unpolarized radiation [19]. Similar to the previous studies mentioned above, the minimum detectable concentration at a depth of 4 cm is about 8 μg/g for a 30 min measurement time [19]. However, one cannot compare this low marker concentration with preclinical experiments, because in a human kidney, the total mass of cisplatin is, even at such low concentrations, effectively higher than in a mouse kidney.

Even though the method based on ^109^Cd has been considered as very effective for a couple of years, it is not capable of measuring low-level lead concentrations as they are present in the general population [20]. With the help of Monte Carlo simulations and experiments using phantoms, Nie et al. [20] designed an improved system and predicted that it would be about three times more sensitive than the conventional system.

In parallel, the construction of dedicated synchrotron light sources providing high-flux, focused, monochromatic, tunable and polarized X-ray pencil-beams, and the development of computer-assisted tomography for imaging, led to a new technique called X-ray fluorescence tomography [21]. The first runs were carried out at the National Synchrotron Light Source at Brookhaven National Laboratories in 1985, using a setup consisting of a monochromatic, focused and collimated X-ray beam, hitting a sample mounted on a goniometer and measuring the emitted characteristic X-ray fluorescence photons with a Si (Li) detector positioned at 90°. Already, those early studies could show that computerized fluorescence tomography of small samples to study elemental distributions of minor and trace elements is practical, and that spatial resolutions on the micrometer-scale are feasible [21]. However, due to the limited access to synchrotron facilities, X-ray fluorescence computed tomography (XFCT) could not be made widely available for experiments, leading to the first attempts of designing bench-top systems [22]. At least one early study [23] concluded that an XFCT system using a special X-ray tube that can produce quasi-monochromatic X-rays would not be practical for human applications in terms of achievable spatial resolution, minimum detectable concentration and scanning time [22]. In [23], a parameter set for medical applications was studied, consisting of a water cylinder with a 30 cm diameter as phantom, gadolinium (Gd), iodine (I) and gold (Au) as marker substances, a 10 mGy of absorbed energy dose, as well as a dedicated detector and collimator geometry. The collimator was considered ideal, meaning that the lamellas were assumed to be infinitely thin and at the same time, perfect radiation absorbers. Likewise, the multiple simulated detector elements were considered as ideal, meaning that real properties such as efficiency and escape peaks were neglected. Different setups (fan-beam or pencil-beam), phantom geometries and sizes, detector angles, as well as excitation energies are studied in simulations and experiments in [23], with the conclusion that the main reasons for the weak performance of XFCT in a clinical scenario stem from the underlying physics, and therefore, cannot be overcome by technological progress on a mid- or long-term time scale. This has long been seen as a show-stopper for translation into clinical applications.

## 3. Translation of X-ray Fluorescence Imaging (XFI) to Clinical Applications

This intrinsic “background problem” [24] arising in large objects can, however, be solved by using X-rays of high brilliance, in combination with advanced spatial and spectral filtering, leading to the necessary reduction of the intrinsic spectral background in X-ray fluorescence imaging (XFI) [24]. It is well-known from several previous studies that this problematic background arises from multiple Compton-scattering events, which lower the photon energy into the signal range of interest. The larger the object, the higher the amount of background photons, as the probability of many sequential Compton-scattering events of each single-incident photon is larger when compared to a preclinical setup with only mouse-sized objects, as the human-sized objects are significantly larger than the mean free path length of the X-ray photons. An advanced spatial and spectral filtering algorithm can be derived from the strong anisotropy of the background and used to minimize intrinsic background contributions to the measured signal, without concomitant signal losses [24]. It was found that the main factors determining the yield of each photon path depend on the total path length and the relative solid angle of a detector’s pixel with respect to the position of the Compton scattering. Taking these main factors into account, the strong anisotropy of the Compton background can be explained and used for a pixel selection algorithm. Based on this finding, in [24], a numerical study demonstrates the practicability of XFI in human-sized objects, as immune cell tracking with a minimum detection limit of 4.4 × 10^5^ cells or 0.86 μg of gold in a cubic volume of 1.78 mm^3^ can be achieved [25].

A comparison of the XFI setup presented in [25] with currently available clinical molecular imaging methods reveals the up- and down-sides of the proposed setup. A clear advantage when comparing XFI to PET/SPECT is the achievable spatial resolution in the mm-range, which is only limited by the size of the incident X-ray beam in XFI, whereas physical limits such as a-collinearity and positron range do exist in PET [26], leading to typical resolution values between 5 and 10 mm [25]. Sensitivity levels for micromolar and nanomolar gold concentrations, as demonstrated in [25], lie in between the achievable levels in magnetic resonance imaging (MRI) and nuclear medicine, similar to image acquisition times, which strongly depend on the area of interest [25]. As any other imaging technique, XFI requires dedicated markers, e.g., gold nanoparticles in the context of human-sized objects, but different to the radioisotopes needed in PET/SPECT, XFI markers do not decay over time, and hence allow longitudinal studies over arbitrarily long time windows (as long as the markers remain in the body). In addition, not only can a single marker element be measured in an XFI scan, but also multiple marker elements simultaneously, called multi-tracking, which is another clear advantage over other imaging modalities as different aspects can be studied in a single scan. A current drawback of the human-sized XFI setup is the high effort required in technology, especially for suitable X-ray detectors and collimators, which become very costly when produced in the size as used in simulations [24,25]. The simulations presented in [25] used an X-ray detector with a big hole on one side to move the voxel phantom inside, which would also be required in a realistic scenario with patients. However, this, in turn, leads to a loss of the sensitive detector area, which is crucial to reach a high sensitivity level. Besides the fact that more simulations are required to determine an optimal detector layout, 4π detectors do not exist as yet, but typical detection areas are rather in the range of a few tens of mm^2^, as used in the demonstration measurements presented in [24]. Therefore, further developments in suitable X-ray detector technology, which is capable of measuring energies and absolute numbers of photons at the same time, are one essential step towards a clinical translation of XFI.

Furthermore, the use of a synchrotron X-ray source as simulated in [25] is impractical for clinical applications, as those machines are huge, expensive and only have very limited access. One potential solution for the translation into clinics is the use of compact, laser-driven X-ray sources, which have become an active field of research in recent years [27,28,29,30]. By using state-of-the-art high-power lasers, it is possible to accelerate electrons to relativistic energies over very short distances due to the creation of highly intense plasma wakefields. Typical laser-wakefield accelerators (LWFA) provide an accelerating field gradient more than 1000 times higher than conventional radiofrequency (RF)-cavity-driven accelerators and are thus much more compact [29]. The concept presented in [29] uses only one single high-power laser beam, which is divided into two synchronized light pulses, of which one pulse drives the LWFA and the other one acts as an undulator by scattering from the relativistic electrons. By using this principle of inverse Compton scattering (also called Thomson scattering in the energy range relevant for medical imaging), quasi-monoenergetic and tunable X-rays can be produced [29]. A dedicated design study for a compact laser-driven source for medical X-ray fluorescence imaging presents an optimization procedure, with the goal to produce X-ray beams of sufficient quality for XFI studies [31]. Several recently published studies demonstrate the basic requirements of the source proposed in [31], such as the stability of a compact laser-plasma accelerator over a typical clinical working day of 8 h, as well as the energies required for producing XFI-suitable incident beams [32]. As high-sensitivity XFI measurements require an incident bandwidth below 15% FWHM [24], which is not fulfilled by typical Thomson sources, an additional electron-focusing device, namely an active plasma lens, has to be implemented in the setup in order to produce tunable X-rays with percent-level bandwidths [33]. The very first XFI demonstration measurements at such a source have shown that the principle works; however, improvements such as an increase in the laser repetition rate and background reduction on the source side are still necessary [34].

## 4. Current Status and Recent Promising Developments of Preclinical XFI Research

Considering that X-ray fluorescence measurements have been seen as impractical for routine in vivo imaging, especially in terms of the scanning time [22], several research groups have instead focused on imaging of smaller objects, mostly in connection with gold nanoparticles (GNPs). In [22], ordinary polychromatic diagnostic energy X-rays from a conventional X-ray tube were used to perform XFCT imaging of GNP-containing objects inside phantoms mimicking tumors/organs within a small animal. While earlier developments of XFCT benchtop settings produced rather disappointing results, adapted approaches using a pencil-beam from polychromatic X-rays could demonstrate the detection of biologically relevant concentrations of GNPs (1–2% by weight) [12] (see Figure 1). Figure 1 demonstrates that the detection sensitivity of XFCT is substantially higher compared to CT. While [12] shows this convincing result, there is no detailed discussion on why this is the case; thus, we wanted to explicate this finding in more detail. CT, or for reasons of simplicity a general X-ray absorption image, relies on a signal difference between neighbored rays, leading to a visible contrast. If we assume, for simplicity, that there are two neighbored rays which traverse a given object of the same thickness for both rays, then a contrast is visible if and only if the difference in (detected) photon counts is significantly larger than the statistical noise of both counts. Such a significant difference can only arise if along the volume of both rays, there is a sufficient difference in the electron density. Therefore, if the tumor size and/or its density difference compared with its surrounding is too small, then no significant signal difference over noise is possible, and the tumor remains invisible, as shown in Figure 1.

The detection sensitivity in X-ray imaging and CT hence solely relies on density differences within the object to be measured. Obviously, the local amount of gold nanoparticles in Figure 1 does not suffice to create a sufficiently increased electron density compared to the surrounding of the tumor. In contrast to X-ray absorption imaging, X-ray fluorescence imaging does not rely on such local density differences, but only on the ability to excite and detect a sufficient number of characteristic fluorescence photons, whereby this number needs to be put in relation to the spectral background in the element-specific signal energy region. Since the attenuation of X-rays in mouse-sized objects does not play a major role (a key advantage over optical fluorescence), XFI only requires a sufficiently large local number of markers at the site of interest, such as a tumor or inflammatory region, but no contrasts with the surroundings. This difference in the corresponding image generation processes explains the much higher degree of sensitivity of XFI compared to X-ray absorption imaging.

Besides pencil-beam approaches, cone beam implementations of XFCT have also been developed, which allow fluorescence signal acquisition, a crucial aspect for making XFCT suitable for in vivo imaging under the practical constraints of the X-ray dose and scan time [12]. In a typical benchtop XFCT setup, as presented in [12], diagnostic X-ray tubes are used in combination with dedicated filters in order to optimize the incident spectrum in terms of quasi-monochromatization and dose; for example, 125 kVp X-rays filtered with 2 mm of tin. With such a setup, a tumor-bearing mouse injected with GNPs was successfully imaged, demonstrating the capabilities of benchtop XFCT under the conditions most relevant to in vivo imaging [12].

In the past years, nanoparticles (NPs) have been emerging as attractive new contrast agents in biomedical imaging due to their capacity for higher sensitivity and for (targeted) drug delivery. In addition, they offer flexible tailoring of both physical and biochemical properties [35]. NPs from different elements were used for XFCT demonstration experiments, e.g., Mo, Gd and Au, reaching different levels of spatial resolution and sensitivity [35].

In a recent proof-of-principle study presented in [35], mice were imaged in vivo in an XFCT setup reaching 100 μm spatial resolution and demonstrating longitudinal imaging by imaging each mouse 5 times (1 h, 1 week, 2 weeks, 5 weeks and 8 weeks post-tail-vein injection of suspension with NPs at a 1% Mo mass fraction). The used setup was a combination of a laboratory pencil-beam arrangement with the sensitive detection of tailored MoO_2_ NPs and real-time monitoring of respiration and body temperature under anesthesia [35]. A liquid-metal-jet microfocus source was coupled to a multilayer Montel mirror, which had a Gaussian reflectance profile centered at 24 keV with a FWHM of about 1.4 keV, hence creating a quasi-monochromatic pencil-beam of 100 μm in diameter [35]. These spectral characteristics are ideal for X-ray fluorescence studies with MoNPs, as their K-absorption edge at 20 keV allows a significant separation from the main Compton-scattering peak at energies above 23 keV [35]. A whole-body projection image with a size of 40 mm × 70 mm took around 15 min, while the acquisition time for a local-region XFCT and CT with 30 projections, each having a size of 40 mm × 20 mm, would take around 1 h to acquire [35]. For the 2D 15-min scan, a radiation dose of 1 mGy was estimated by means of Monte Carlo simulations, using the same imaging geometry as in the experiment and the voxelized digital mouse phantom DIGIMOUSE [36] as the simulated object [35]. However, the XFCT mode required a dose of 22 mGy.

The relative clearance of the measured whole-body signal over time could be correlated to the clearance of the injected nanoparticles, reaching signals close to the background level after 8 weeks [35]. It must be noted here that no quantitative conclusions could be drawn from the full-body projection images since effects such as self-absorption of fluorescence photons can only be modeled from tomographic data [35]. Therefore, additional in vivo XFCT and CT scans were acquired in [35] with the liver as the region of interest due to the major accumulation of NPs observed in that organ, which were then analyzed with an iterative XFCT reconstruction algorithm that allows for quantitative determination of NP concentrations. The detection limit of the imaging system was estimated at 0.05 mg/mL of Mo, but it is noted in [35] that this limit can be linearly improved with an increased pixel exposure time which, however, implies a higher radiation dose and longer scan times. A drawback of this achievement is that no heavier elements than Mo can be imaged due to the current limitation in photon energy of the liquid metal jet source used.

Besides the use of pencil-beam setups with near-monochromatic incident radiation, in vivo biodistribution measurements of gold nanoparticles (GNPs) have also been demonstrated with polychromatic fan-beam X-rays [37]. The combination of a transmission CT detector installed in an existing pinhole XRF imaging system using a two-dimensional cadmium zinc telluride (CZT) camera allows to acquire functional and anatomic information on the same platform [38]. The pinhole XRF system used in [38] comprised a tungsten fan-beam collimator, a lead pinhole collimator and a CZT camera, reaching a spatial resolution of 4.4 mm. Due to the different optimal energy spectra for XRF and CT imaging (high energies above the K-edge of GNPs for XRF and low energies to produce sufficient contrast on CT), images had to be sequentially acquired on the same platform [38]. Nevertheless, the use of a 2D array detector reduces the excessive imaging acquisition time and radiation dose due to fact that 2D XRF images can be directly acquired without image reconstruction [38]. The radiation dose delivered in the dual-imaging setup was 59.1 mGy for the XRF images and an additional 321.7 mGy for the CT image acquisition, which should be reduced by optimizing the X-ray tube parameters, the filter material used, as well as the scanning procedure, in general [38]. Nevertheless, the detection limit of 0.01 wt.% needs to be further improved, e.g., by replacing the CZT detector used with pixelated detectors of better energy resolution and a higher maximum count rate performance [38].

## 5. Conclusions

Since the very first applications in the late 1970s, the method of X-ray fluorescence imaging has made substantial improvements, especially regarding the achievable minimum detection sensitivity and its usage in different application areas. While the first studies mainly used radioactive isotopes and special geometric configurations, setups used nowadays can either be realized at synchrotrons or conventional X-ray sources, where the applied beam diameters and especially the radiation dose can be monitored and controlled with much higher precision.

Even though the suitability of X-ray fluorescence imaging for preclinical and clinical applications was considered unlikely in the early 2000s, there has been tremendous progress to improve the modality in recent years. On the one hand, compact setups have been developed in order to enable measurements at existing X-ray systems, while on the other hand, different strategies have evolved to overcome intrinsic background limitations. The use of dedicated filters, collimators, pinholes or pixelated detectors nowadays allows the detection of low marker concentrations even in large objects, which clearly paves the way towards future clinical applications. A current limitation lies in the fact that measurements of the highest sensitivity can only be performed at synchrotron beamlines, which in turn strongly limits the potential applications. Therefore, it is essential to further develop compact systems which will allow usage of the modality in laboratory and clinical environments.

Different strategies for such compact X-ray systems have already been demonstrated, in which most combine XFI and CT imaging in order to gain functional and anatomical information in one measurement. The main challenge of these systems currently lies in the fact that the incident radiation from a conventional X-ray tube needs to be focused and monochromatized in order to achieve measurements of the highest spatial resolution and detection sensitivity. One promising solution is the use of dedicated X-ray optics, which allow to focus a certain X-ray energy of interest; however, their efficiency needs to be improved to allow for measurements of acceptable imaging times and radiation doses [39].

Overall, the application areas of XFI are manifold, reaching from measurements of elemental distributions in non-destructive testing, to uptake studies of certain entities into single cells, to different applications in medical imaging, such as biodistribution studies of new medical drug compounds or tumor localization measurements with the highest precision. Thus, XFI bears the convincing potential to complement other already well-established molecular imaging methods in areas where XFI offers unprecedented data, e.g., the simultaneous in vivo tracking of different immune cell subtypes in preclinical research, with both high spatial resolution and sensitivity.

## Figures and Tables

**Figure 1 ijms-24-10990-f001:**
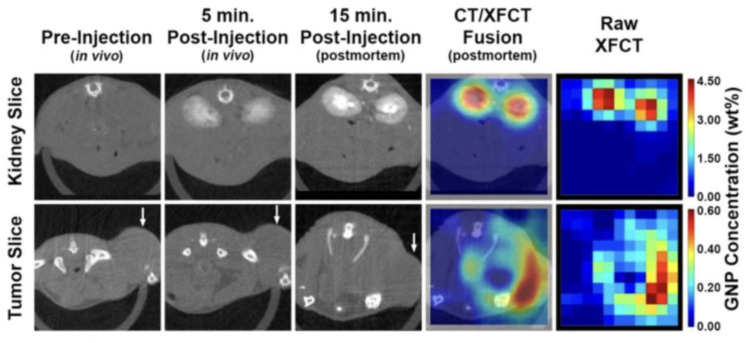
Taken from [12]: CT and XFCT maps from a scan of a tumor-bearing mouse. One can clearly see that the CT maps do not show any gold nanoparticle locations, in contrast to the XFCT maps, thus demonstrating the superior sensitivity of the latter. White arrows indicate the levels of the kidneys and tumor.

## Data Availability

Data will be provided by the authors upon reasonable request.

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
