# Peer review of "Review of Development and Recent Advances in Biomedical X-ray Fluorescence Imaging"

_ijms, 2023, doi:10.3390/ijms241310990_

Round 1

Reviewer 1 Report

Reviewer

Initial comments

Figure 1. (taken from [27]): CT- and XFCT-maps from a scan of a tumor-bearing mouse. One can clearly see that the CT-maps do not show any gold nanoparticle locations, in contrast to the XFCT- map, thus demonstrating the superior sensitivity of the latter compared to the first.

(X-ray fluorescence computed  tomography (XFCT)

       This is an image examination paper, and Figure 1 clearly shows the importance of images for diagnostic purposes.

       So, this paper of “Review of Development and Recent Advances in Biomedical X- Ray Fluorescence Imaging” it is very important and deserves to be published.

Review

Abbreviations – The paper is very extensive and has many acronyms, and a list of abbreviations would be advisable for a better understanding of the text.

Title

Review of Development and Recent Advances in Biomedical X- Ray Fluorescence Imaging

Comment:

It is suitable

Abstract:

Comment:

It is suitable

1. Introduction  

 Comment:

It is suitable

2. History and developments of X-ray fluorescence measurement techniques

Comment:

It is suitable

3. Translation of X-ray fluorescence imaging (XFI) to clinical applications

Comment:

It is suitable

4. Current status of preclinical XFI-research

Comment:

It is suitable

5. Conclusions

Comment:

It is suitable

References

Comment:

It is suitable

Thank you

Author Response

Dear reviewer,

thank you very much for your very positive feedback to our manuscript. You mentioned that the paper does have many acronyms and that a list of abbreviations would be advisable for a better understanding of the text. This is a very valuable comment and we did add the following list to our manuscript:

XFI – X-ray Fluorescence Imaging

XRF – X-ray Fluorescence

XAS – X-ray Absorption

XES – X-ray Emission

CT – Computed Tomography

PET – Positron Emission Tomography

XFCT – X-ray Fluorescence Computed Tomography

SPECT – Single Photon Emission Computed Tomography

LWFA – Laser-Wakefield Acceleration

RF – Radio Frequency

FWHM – Full Width at Half Maximum

NPs - Nanoparticles

GNPs – Gold Nanoparticles

CZT – Cadmium Zinc Telluride

We hope that this list provides a better understanding of the text and thank you again for the positive feedback!

Reviewer 2 Report

1.      The introduction does not clearly state the objectives and scope of the review article. It is important to define the specific goals and purpose of the article to provide a clear roadmap for the reader. Additionally, the overall structure of the article needs to be more organized and coherent.

2.       The introduction briefly mentions the use of X-rays for non-invasive imaging but lacks sufficient context and historical background. It is important to provide a more comprehensive overview of the history and evolution of X-ray imaging techniques to set the stage for discussing X-ray fluorescence imaging (XFI).

3.      The claims made in the introduction, such as "several well-established methods," "typical applications," and "intrinsic spectral background issues," lack proper citations or references to support these statements. It is essential to back up these claims with relevant literature and studies to establish credibility and provide a foundation for the review.

4.      The description of XFI is brief and lacks details. It is important to provide a more thorough explanation of the technique, including its principles, methodologies, and advantages over other imaging methods. This will help readers understand the significance of XFI in preclinical and clinical applications.

5.      The article mentions "recent promising developments" without providing specific details or examples. It is crucial to elaborate on these advancements, such as novel techniques, studies, or technological breakthroughs, and their impact on the field of XFI. This will provide readers with valuable insights into the current state of the art.

6.      The article briefly mentions the potential for clinical translation of XFI without delving into the specific challenges, strategies, or advancements required for successful implementation. It is important to provide a more comprehensive and detailed analysis of the steps needed to overcome limitations and facilitate the adoption of XFI in clinical settings.

7.      The article does not have a clear conclusion that summarizes the main points and key findings. A well-crafted conclusion is essential to wrap up the review, reinforce the main takeaways, and provide a sense of closure for the reader.

8.      The article lacks proper references to support the claims and statements made throughout. It is essential to provide relevant citations from reputable sources to substantiate the information presented and enable readers to explore the topic further.

9. Overall, the article requires significant revisions and improvements in terms of organization, clarity, depth of content, and scholarly rigor. Considering these factors, it is advisable to reject the article in its current form and request a thorough revision addressing the outlined concerns.

The article contains grammatical errors, awkward sentence structures, and lacks clarity in certain sections. It is necessary to thoroughly proofread the entire article to improve readability and ensure that the language is concise, precise, and coherent

Author Response

Dear reviewer,

thank you very much for your comments which definitely helped to improve our manuscript. Please find our answers to your comments below.

  1. The introduction does not clearly state the objectives and scope of the review article. It is important to define the specific goals and purpose of the article to provide a clear roadmap for the reader. Additionally, the overall structure of the article needs to be more organized and coherent.

Thank you for this important comment. In order to provide a clear roadmap for the reader and give an impression already at the very beginning, we added the following paragraph to the manuscript:

“This review article aims to give an historical overview of the development of X-ray fluorescence measurement techniques and to summarize the recent developments in the field. A detailed description of different experimental setups, used X-ray sources and the thereby achievable detection limits in various application areas are presented. Besides the current status in preclinical research, a translation of XFI to clinical applications is presented.”

  1.  The introduction briefly mentions the use of X-rays for non-invasive imaging but lacks sufficient context and historical background. It is important to provide a more comprehensive overview of the history and evolution of X-ray imaging techniques to set the stage for discussing X-ray fluorescence imaging (XFI).

We added the following paragraph about the development of X-ray imaging techniques to motivate the usage of X-ray fluorescence imaging:

“Besides techniques and applications using X-rays to study smallest structures, also radiological techniques were gradually improved over the years such as the nowadays widespread computer tomography [1]. X-ray computed tomography (CT) consists of measuring attenuation profiles of transverse slices of patients from many different angular positions by using a fan or cone beam from an X-ray tube in conjunction with a detector array traveling on a circular path opposite to the X-ray source around a patient [6]. Nowadays, CT scans are typically used to diagnose many diseases such as various types of cancers, heart diseases like myocardial disease and analyses of the liver or pancreas of patients [7]. Since the 1970s and 1980s, the speed of image acquisition has improved substantially and modern CT scanners are capable of imaging patients in a matter of seconds or less [8]. A major drawback of CT is that large masses within the gastrointestinal tract may not be visible during scans and more sophisticated methods like dual-energy CT are required to differentiate materials with the same attenuation at a certain energy for better lesion depiction [7,9].

  1. The claims made in the introduction, such as "several well-established methods," "typical applications," and "intrinsic spectral background issues," lack proper citations or references to support these statements. It is essential to back up these claims with relevant literature and studies to establish credibility and provide a foundation for the review.

We extended the Abstract of our manuscript to support our statements with a bit more details. The corresponding references are found later in the text since citations are not allowed in the Abstract itself. Please find the extended version of our Abstract below:

“The use of X-rays for non-invasive imaging already has a long history which resulted in several well-established methods in preclinical as well as clinical applications such as tomographic imaging or computed tomography. While projection radiography provides anatomical information, X-ray fluorescence analysis allows quantitative mapping of different elements in samples of interest. Typical applications so far comprise the identification and quantification of different elements and are mostly located in material sciences, archeology and environmental sciences, whereas the use of the technique in life science has been strongly limited by intrinsic spectral background issues arising in larger objects so far. This background arises from multiple Compton scattering events in the objects of interest and strongly limits the achievable minimum detectable marker concentrations. Here, we review the history and report on the recent promising developments of X-ray fluorescence imaging (XFI) in preclinical applications and also give an outlook on the clinical translation of the technique, which can be realized by reducing the above-mentioned intrinsic background with dedicated algorithms and by novel X-ray sources.”

  1. The description of XFI is brief and lacks details. It is important to provide a more thorough explanation of the technique, including its principles, methodologies, and advantages over other imaging methods. This will help readers understand the significance of XFI in preclinical and clinical applications.

In order to explain the modality of XFI in more detail and to compare it to other imaging modalities, we added the following paragraph to our manuscript (where we also added an explanation of the used abbreviations that was originally written later in the text):

“The abbreviations XRF and XFI are often used interchangeably, hence it is noted here that both versions will be used in the following, where the choice depends on the use of the cited publication. Different to other molecular imaging methods, the spatial resolution in XFI only depends on the size of the applied X-ray beam and does not face any physical limitations [10]. In order to make entities of interest visible with XFI, dedicated markers have to be coupled to them, such as for example metallic nanoparticles or molecular tracers, for example iodine atoms. Given the fact that these markers do not decay over time, longitudinal studies are possible to study the biodistribution of labelled entities over long time spans in one and the same object. Furthermore, several entities can be tracked simultaneously by using different marker elements and measurements on completely different size scales, starting from full-body in vivo scans of small animals and going down to measurements at the singe-cell level are feasible with XFI [11,12]. Especially the last two aspects are an advantage compared to other commonly used imaging methods such as Positron Emission Tomography (PET) where only single markers can be imaged over limited time spans due to the half-life of only 110 minutes of 18F, the workhorse of PET [13].

Compared to XFI, the sensitivity of CT is much reduced, for instance, a tumor marked with gold nanoparticles in a tumor-bearing mouse model is undetectable for CT while XFI can clearly locate it [12], as described in detail below. The main reason for this difference in detection sensitivity is the fact that the contrast in CT arises due to a difference in photon counts in forward (transmission) direction, while XFI is a spectroscopic method, where the fluorescence photons are emitted isotropically and the detection limit only depends on the spectral background in the signal region determined by multiple Compton scattering.”

  1. The article mentions "recent promising developments" without providing specific details or examples. It is crucial to elaborate on these advancements, such as novel techniques, studies, or technological breakthroughs, and their impact on the field of XFI. This will provide readers with valuable insights into the current state of the art.

We modified the title of chapter 4 to emphasize that the recent developments regarding novel imaging setups, used X-ray sources as well as radiation detectors are presented there.

  1. The article briefly mentions the potential for clinical translation of XFI without delving into the specific challenges, strategies, or advancements required for successful implementation. It is important to provide a more comprehensive and detailed analysis of the steps needed to overcome limitations and facilitate the adoption of XFI in clinical settings.

We added the following paragraph to chapter 3 to explain the challenges related to a clinical translation of the method in more detail:

“The simulations presented in [25] used an X-ray detector with a big hole on one side to move the voxel phantom inside, which would also be required in a realistic scenario with patients. However, this in turn leads to a loss of sensitive detector area which is crucial to reach a high sensitivity level. Besides the fact that more simulations are required to determine an optimal detector layout, 4 detectors do not exist as of yet, but typical detection areas are rather in the range of a few tens of mm2, as used in the demonstration measurements presented in [24]. Therefore, further developments in suitable X-ray detector technology which is capable of measuring energies and absolute numbers of photons at the same time, are one essential step towards a clinical translation of XFI.”

  1. The article does not have a clear conclusion that summarizes the main points and key findings. A well-crafted conclusion is essential to wrap up the review, reinforce the main takeaways, and provide a sense of closure for the reader.

Thank you again for raising this important point. We added the following paragraphs to make the conclusion clearer for the reader:

“Since the very first applications in the late 1970s, the method of X-ray fluorescence imaging has made substantial improvements especially regarding the achievable minimum detection sensitivity and its usage in different application areas. While the first studies mainly used radioactive isotopes and special geometric configurations, setups used nowadays can either be realized at synchrotrons or conventional X-ray sources where the applied beam diameters and especially the radiation dose can be monitored and controlled with much higher precision.

Different strategies for such compact X-ray systems have already been demonstrated, in which most combine XFI and CT imaging in order to gain functional and anatomical information in one measurement. The main challenge of those systems currently lies in the fact that the incident radiation from a conventional X-ray tube has to be focused and monochromatized in order to achieve measurements of highest spatial resolution and detection sensitivity. One promising solution is the use of dedicated X-ray optics which allow to focus a certain X-ray energy of interest; however, their efficiency needs to be improved to allow for measurements of acceptable imaging times and radiation dose [38].

Overall, the application areas of XFI are manifold and reach from measurements of elemental distributions in non-destructive testing, over uptake studies of certain entities into single cells, up to different applications in medical imaging such as biodistribution studies of new medical drug compounds or tumor localization measurements with highest precision. Thus, XFI bears the convincing potential to complement other already well-established molecular imaging methods in areas where XFI offers unprecedented data, like e.g., the simultaneous in vivo tracking of different immune cell subtypes in preclinical research with both high spatial resolution and sensitivity.”

  1. The article lacks proper references to support the claims and statements made throughout. It is essential to provide relevant citations from reputable sources to substantiate the information presented and enable readers to explore the topic further.

We added several new and more appropriate references from reputable sources to better support our claims and statements throughout the whole manuscript.

  1. Overall, the article requires significant revisions and improvements in terms of organization, clarity, depth of content, and scholarly rigor. Considering these factors, it is advisable to reject the article in its current form and request a thorough revision addressing the outlined concerns.

We hope that our thorough revision does address all the major concerns raised in the comments and we want to thank the reviewer again for the valuable feedback which really helped to improve our manuscript.

Round 2

Reviewer 2 Report

Accepted

 Minor editing of the English language required